# Application of a variational autoencoder for clustering and analyzing *in situ* articular cartilage cellular response to mechanical stimuli

Jingyang Zheng[1], Han Kheng Teoh[1]*, Michelle L. Delco[2], Lawrence J. Bonassar[3,4], Itai Cohen[1]

1 Department of Physics, Cornell University, Ithaca, NY, United States of America, 2 College of Veterinary Medicine, Cornell University, Ithaca, NY, United States of America, 3 Meinig School of Biomedical Engineering, Cornell University, Ithaca, NY, United States of America, 4 Sibley School of Mechanical and Aerospace Engineering, Cornell University, Ithaca, NY, United States of America

☯ These authors contributed equally to this work.
* ht452@cornell.edu

**Data Availability Statement:** The data and code can be found in the Zenodo repository at: https://doi.org/10.5281/zenodo.10565588.

## Abstract

In various biological systems, analyzing how cell behaviors are coordinated over time would enable a deeper understanding of tissue-scale response to physiologic or superphysiologic stimuli. Such data is necessary for establishing both normal tissue function and the sequence of events after injury that lead to chronic disease. However, collecting and analyzing these large datasets presents a challenge—such systems are time-consuming to process, and the overwhelming scale of data makes it difficult to parse overall behaviors. This problem calls for an analysis technique that can quickly provide an overview of the groups present in the entire system and also produce meaningful categorization of cell behaviors. Here, we demonstrate the application of an unsupervised method—the Variational Autoencoder (VAE)—to learn the features of cells in cartilage tissue after impact-induced injury and identify meaningful clusters of chondrocyte behavior. This technique quickly generated new insights into the spatial distribution of specific cell behavior phenotypes and connected specific peracute calcium signaling timeseries with long term cellular outcomes, demonstrating the value of the VAE technique.

## Introduction

Exploring how cells coordinate their behaviors in response to stimuli is important for understanding tissue function in health and disease. Tissues are complex systems where many factors such as spatial location and mechanical stimuli can affect mechanical and biochemical signal transduction between cells and dictate the subsequent cellular response [1]. Methods have been developed to approach this problem through imaging individual cell behaviors over time [2–7] or taking snapshots of pooled cell populations [8–10]. However, fully addressing this problem can be difficult. In order to make specific conclusions about how coordinated cellular behaviors are affected by external factors, the spatial and temporal behaviors for each of

**Funding:** Funding: The work was supported by the NIH National Institute of Arthritis and Musculoskeletal and Skin Diseases, Contract: K08AR068470, R03AR075929, and The Harry M. Zweig Fund for Equine Research. This work was also supported by the NIH National Institute of Neurological Disorders and Stroke. Contract: R01NS116595. Additionally, this work was supported by the National Science Foundation grants DMR-1807602, CMMI 1927197, and BMMB-1536463. Lastly, this work made use of the Cornell Center for Materials Research Shared Facilities which are supported through the NSF MRSEC program (DMR-1719875).

**Competing interests:** The authors have declared that no competing interests exist.

thousands of cells within a tissue must be collected, processed, and interpreted for many iterations of experiments. Experiments with different stimuli must then be compared in order to determine how behaviors may change. This process amounts to an overwhelming amount of data for analysis using well-established methods.

Recently, we developed a technique that combines real-time *in situ* imaging of cartilage tissue during impact with supervised machine learning techniques to establish and probe specific behaviors within the system, called STRAINS [11]. The STRAINS method enables a detailed spatiotemporal analysis of individual cell behaviors and the classification of specific cell responses and phenotypes. However, this process requires extensive manual classification, which is initially time consuming for large datasets and may fail to distinguish subtle differences between cells due to human error.

To overcome these limitations, we propose using the variational autoencoder (VAE), an unsupervised ML method, for simultaneous analysis of thousands of time series. The VAE, a probabilistic generative neural network, iteratively learns to reproduce input data accurately and map it to a latent space through an encoder, a decoder, and a loss function. The latent space of the VAE is connected (two points in the space that are close together give similar decoded results) and complete (all points in the latent space give meaningful information upon decoding), preventing overfitting and enabling the generation of new data. VAEs, with their ability to process large amounts of data, find applications in diverse fields, from reconstructing complexities in many-body physics [12] to anomaly detection in industrial robots [13].

In biomedical research, Variational Autoencoders (VAEs) have found widespread application, particularly as a diagnostic tool for image classification in MRIs and other medical imaging modalities [14]. They have demonstrated efficacy in tasks such as tumor classification, image segmentation [15, 16], multi-omics data integration, and even in the design of molecules and proteins [17]. The processing capability of VAEs led to its use in single-cell analysis techniques, focusing on specific biomarkers in individual cells [18] and single-cell transcriptome profiling [19]. Moreover, VAEs have proven instrumental in handling time-dependent biological signals within spectrographic data. This is evident in their application to cluster and analyze the vocalizations of songbirds and mice, showcasing their versatility in understanding complex biological phenomena [20–22].

This paper aims to demonstrate the utility of a VAE for analyzing large-scale cellular response in articular cartilage to mechanical stimuli—a process traditionally requiring a time-consuming analysis pipeline. We showcase the VAE's ability to accurately reconstruct temporal features of cellular behavior and leverage latent features for phenotype identification immediately after injury. Additionally, we highlight the VAE's role in hypothesis generation and validation, providing a comprehensive understanding of cell response post-injury.

## Methods

### Impacting articular cartilage tissue

Sterilely-dissected 6mm cylindrical explants of articular cartilage were cultured and stained for calcium concentration (Calbryte 520AM), mitochondrial polarity (tetramethylrhodamine, TMRM), and nuclear membrane permeability (Sytox Blue). Selected stains were chosen to reflect relevant parameters for this study but can be readily modified. Following dissection, samples were bisected and affixed to the back plate of a confocal-mounted impactor, submerged in a bath of PBS (Dulbecco's Phosphate Buffered Saline) and Sytox Blue stain. This allowed for the tracking of cell death dynamics during the experiment. One-half of each sample served as the impacted group, while the other half acted as a control. The impactor delivers

an energy-controlled impact using a spring-loaded piston, producing a peak stress of $\sim 1$MPa over 5-10ms and replicating a superphysiologic loading rate known to induce tissue damage in cartilage.

Throughout the experiment, the impactor remained mounted on the stage of an inverted spinning disk confocal microscope (3i Marianas). Utilizing a 10x objective, imaging covered the region of impact, the lateral and sub-impact regions, and two corresponding depth regions on the control sample. Each imaged region measured 660μm x 660μm (512 x 512 pixels). During the impact and the subsequent minute, the region of impact was imaged at approximately 40 frames per second. For longer-term observation, continuous imaging was maintained at all sites, with roughly 12 seconds between frames. The imaging depth was set at approximately 30 μm below the cartilage cut surface to avoid capturing chondrocytes damaged during sample handling.

### Extracting time series data of chondrocyte behavior

Cells in the images were tracked using a modified version of the Crocker and Grier algorithm [23]. As described in previous work, the color channels were summed, and linear interpolation and static extrapolation were used to obtain cell centroid locations [11]. These centroid values served as the basis for defining small sub-cellular regions over which fluorescence values were averaged, producing fluorescence traces for all three channels per cell. To mitigate high-frequency noise, an additional moving-average smoothing step was incorporated, with a carefully chosen window size that avoids affecting features within the time series.

During and after cartilage impact, cell-localized stains leaked into the extracellular matrix and eventually dissipated. This led to a localized increase in background intensity for certain video frames, resulting in a non-uniform background spatially and temporally. To address this, additional background subtraction was implemented. A grid of 8x8 subsets was generated within each image, and the mean of the twenty lowest non-zero pixel values within each subset was subtracted. Subsequently, the time-series data for each cell was smoothed with a window size of 10 and re-sampled to 750 time points to match the input dimension of a VAE.

### VAE structure

The encoder is composed of three 1D convolution layers featuring a kernel size of 3, a stride of 2, and a padding size of 1. The input and output channels are configured as follows: i) (3, 4), ii) (4, 8), and iii) (8, 16). Following the convolutional layers is a fully connected layer with input and output channels set at (1504, 256). The resulting output is then directed into two branches of fully connected layers, where each branch further comprises two fully connected layers with input and output channels specified as i) (256, 128) and ii) (128, 32). These two branches are responsible for generating the latent means and variance vectors. All encoder layers employ a tanh activation function except for the final output layer. Additionally, batch normalization is applied to the data as it passes through the convolutional layers.

The decoder is comprised of three fully connected layers, where the input and output channels are defined as i) (32, 128), ii) (128, 256), and iii) (256, 1504), respectively. The output from the last fully connected layer is reshaped to attain dimensions of ($B$, 16, 94), with $B$ representing the batch size. Subsequently, this reshaped output is passed through three transposed convolution layers, each featuring a kernel size of 3, a stride of 2, and a padding size of 1. The output padding parameters are adjusted to ensure that the final output of the transposed convolutional layers restores the original time series dimension. Similar to the encoder architecture, a tanh activation function is applied throughout, with the exception of the final output

layer. Moreover, latent vectors undergo batch normalization as they pass through the transposed convolutional layers.

To train the Variational Autoencoder (VAE), the time series data $x$ is passed through the encoder, yielding latent means $\mu_\phi(x)$ and latent diagonal covariance $\sigma_\phi(x)$ vectors that define a 32-dimensional normal distribution, denoted as $q_\phi(z|x) = \mathcal{N}(\mu_\phi(x), \sigma_\phi(x))$, where $\phi$ denotes the weights and biases of the encoder. Subsequently, a sample $z$ is drawn from the 32-dimensional Gaussian and propagated through the decoder, parameterized by weights and biases $\theta$, to produce a reconstructed intensity profile. The optimization of the encoder and decoder's weights and biases $\phi$, $\theta$ involves maximizing the log-likelihood of generating real data, log $p_\theta(x)$, while simultaneously minimizing the information loss when the encoder distribution $q_\phi(z|x)$ is used to approximate the true posterior distribution, $p_\theta(z|x)$. The information loss can be quantified via the KL divergence, $D_{\mathrm{KL}}(q_\phi(z|x)|p_\theta(z|x))$. The VAE loss function, denoted as $\mathcal{L}_{\mathrm{VAE}}$, is formulated as:

$$\mathcal{L}_{\mathrm{VAE}} = \log p_\theta(x) - D_{\mathrm{KL}}(q_\phi(z|x)|p_\theta(z|x)). \qquad (1)$$

In practice, this optimization is achieved by minimizing the evidence lower bound objective (ELBO),

$$\mathcal{L}_{\mathrm{ELBO}} = -\mathbb{E}_{z \sim q_\phi(z|x)} \log p_\theta(x|z) + D_{\mathrm{KL}}(q_\phi(z|x)|p_\theta(z)) \qquad (2)$$

which consists of two terms: 1) expected log-likelihood of the decoder distribution, which minimizes the prediction error of the reconstruction, and 2) a regularization term that seeks to minimize the difference between the encoder distribution $q_\theta(z|x)$ and the prior distribution, $p(z)$. Here, we assume the prior distribution over the latent features $z$ to be unit Gaussian, $\mathcal{N}(0, \mathbb{I})$.

## Training details

The VAE was implemented using PyTorch (v1.1.0) [24], where we set the latent dimension $d$ to be 32. We randomly selected 80% of the data as our training set and the remaining 20% as our validation set. We trained the VAE for a maximum of 100 epochs with a batch size of 32 using the Adam optimizer with a learning rate 0.001. To make the network learn robust representations [25], we also injected a unit Gaussian noise $\mathcal{N}(0, 0.01\mathbb{I})$ into the cell data in our training process.

## Quantifying accuracy of reconstructed time series

To characterize VAE's ability to reconstruct the time series, we performed Seasonal-Trend decomposition using LOESS (STL) (statsmodels package on Python [26]) on the reconstructed cellular time series to examine its ability to learn temporal features of short and long time scales. Here, we chose the length of the seasonal smoother to be 35, which we found sufficient to isolate the transient cellular signals. We computed the normalized difference between the original and reconstructed time series on the two time scales, whose distribution is as shown in Fig 1.

## Principal component analysis

To learn how cellular responses are encoded within the latent representation, we performed principal component analysis (PCA) on the latent mean vectors $\vec{\mu}_z$ generated via the VAE encoder. PCA reduces the dimensionality of our data by projecting the data onto a new set of axes, with each subsequent axis capturing less variation. This allows us to determine the first

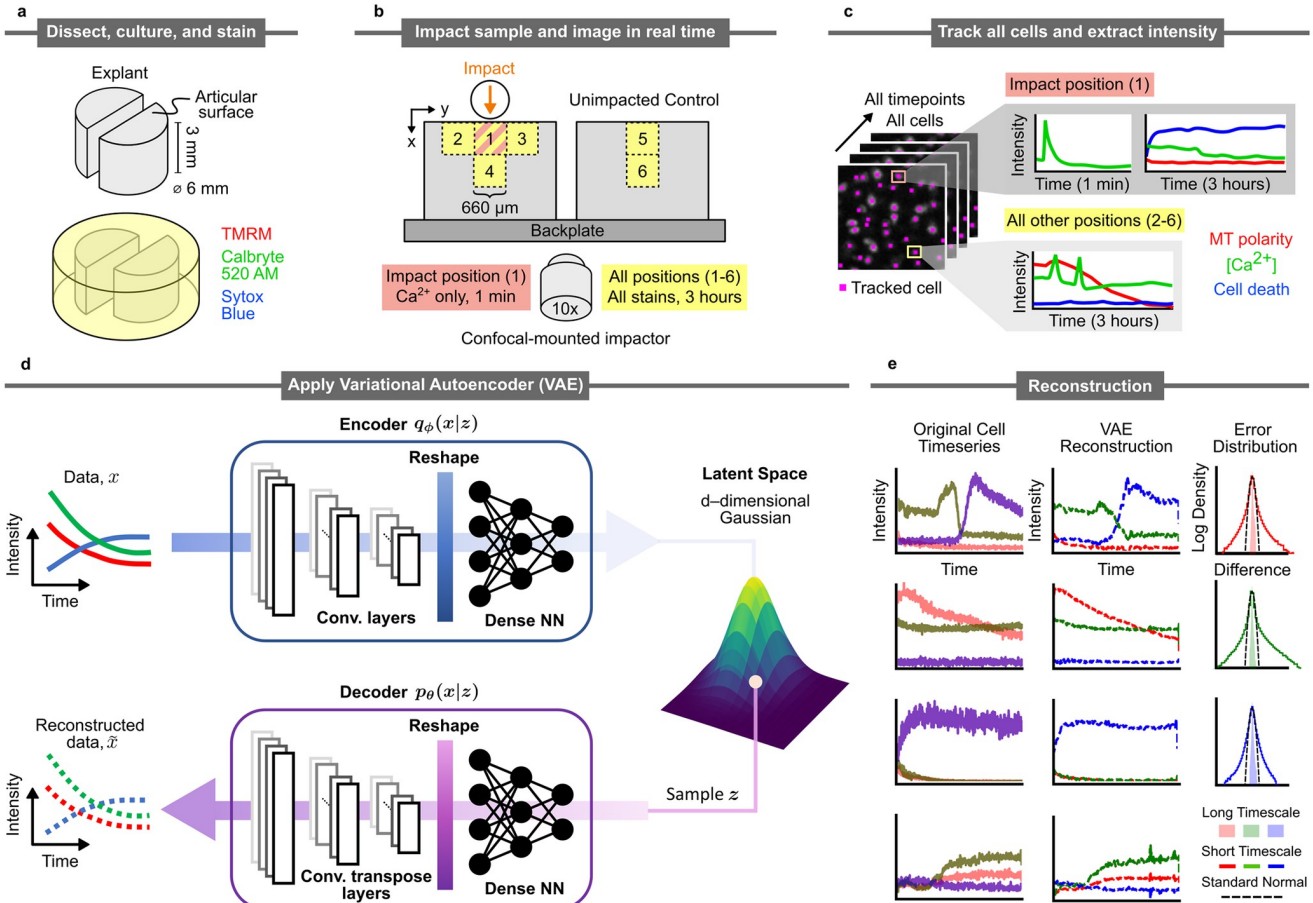

**Fig 1. Sample processing, imaging, tracking, and VAE structure.** a) Biopsy punches of condylar articular cartilage were collected, bisected, and halved. Hemicylinders were stained for calcium concentration, mitochondrial polarization, and nuclear membrane permeability (cell death) via Calbryte 520 AM, Sytox Blue, and Tetramethylrhodamine, respectively. b) Paired hemicylinders were glued side-by-side to the back plate of a confocal-mounted impactor. One hemicylinder was impacted (site 1), while the other served as control. Imaging occurred during impact at site 1 and after impact at sites 1-6. c) Individual cells were tracked through all time points. Stain intensities were extracted and manually sorted (for one sample to provide a comparison baseline for VAE analysis). d) Structure of the Variational Autoencoder (VAE). The encoder consists of three 1D convolution layers and three fully connected layers that output the mean and diagonal covariance vectors for a given time series data, $x$. The decoder has the same architecture as the encoder, with the exception that the 1D convolution layers are replaced with 1D transposed convolution layers. A sample $z$ is drawn from the $d$-dimensional Gaussian and sent through the decoder to obtain a reconstructed intensity profile, $\hat{x}$ e) Example reconstruction of cell intensity data. The left column represents the input data, the middle column represents the reconstructed time series, and the right column shows the error distributions for the reconstructed stain intensities for short and long time scales compared to a standard normal. Timescale decomposition was done with the STL technique (see Methods).

few orthonormal axes $\vec{e}_i$ within the latent space that captures the most variation of the cell's latent mean vectors,i.e $\vec{\mu}_z = \approx \alpha_1 \vec{e}_1 + \alpha_2 \vec{e}_2 + \ldots$, where $\alpha_i = \langle \vec{z} \cdot \vec{e}_i \rangle$ is the dot product between $\vec{\mu}_z$ and $\vec{e}_i$. We sampled along $\vec{e}i$ by adjusting $\alpha_i$ while keeping the rest of the $\alpha_j$ values at zero. The cellular profile encoded along $\vec{e}_i$ was then reconstructed by feeding these sampled latent representations through the decoder.

## Hierarchical clustering

We used the agglomerative hierarchical clustering technique to cluster the cell data in an unsupervised manner. This technique works by treating each cell data as an individual cluster. Similar clusters are merged at each iteration until predefined clusters are formed. We performed

the clustering using the latent representation of cell data obtained from the VAE. As the clustering technique requires a metric to quantify the differences between latent representations, we thus took an information geometric approach in quantifying the dissimilarity between distributions. As the latent representation consisting of a pair of 32-dimensional vectors (mean and diagonal variance) that describe a 32-dimensional Gaussian distribution, we used symmetrized Kullback-Liebler divergence (sKL), a canonical distance measure that quantifies the dissimilarity between two distributions that belongs to the exponential family [27–29].

## Results

To generate training data for VAE, an articular cartilage sample was sterilely bisected and stained for calcium concentration, mitochondrial polarity, and nuclear membrane permeability, as shown in Fig 1a. The stained cartilage sample was loaded onto a confocal-mounted impactor, side-by-side with a control (Fig 1b). The impact site was imaged at 40 frames per second for one minute during and following impact to capture the peracute calcium signaling response. The impact site and surrounding regions were imaged every 10 seconds for three hours after impact for calcium concentration, mitochondrial polarity, and nuclear membrane permeability to capture the longer-term cell behaviors following impact. All of the cells at multiple locations around the impact site and similar sites on the control sample were tracked over time, and the calcium concentration, mitochondrial polarity, and nuclear membrane permeability data were extracted as time series (Fig 1c). This process was repeated for three articular cartilage samples, each of which contained more than 8000 tracked cells.

We then trained a VAE (Fig 1d) on these time series for 100 epochs to learn compressed representations of the cell data. We found that the VAE can accurately reproduce long-timescale stain intensities (Fig 1e left-middle). However, due to the random temporal occurrence of calcium transients, the VAE was unable to accurately reconstruct those features, as illustrated in the long tail error distribution (Fig 1e right). We then utilized the trained VAE to obtain useful insights into the cell behaviors in the system. This was done in three ways: 1) we explored the learned cellular behaviors by probing the variation of the cellular profile encoded along the dominant components of the latent space, 2) we leveraged the learned latent features in clustering cellular profiles to produce meaningful categorizations of cell behaviors, and 3) we demonstrated the utility of VAE in generating and validating hypotheses from the grouped cell behaviors.

### Principal components of the VAE generate valuable insights

To understand how the VAE learned the cellular profiles, we performed Principal Component Analysis (PCA) on the mean latent vectors generated by the VAE. PCA allows one to examine the dominant directions—principal components (PCs)—of the latent space used in capturing the variation of cellular behavior. For our data, 75% of the cumulative explained variance can be accounted for in the first four PCs and 90% in the first eight PCs (Fig 2). Utilizing the VAE decoder, we sampled the cellular profile encoded along the PCs by constructing a synthetic latent feature $z = \sum_i \alpha_i PC_i$ where $\alpha_i$ is a coefficient emphasizing the $PC_i$. Examples of Principal Components (PCs) can be seen in Fig 3 for three different datasets. Fig 3a plots the first twelve PCs as we varied one PC from $\alpha = -3$ to $\alpha = +3$ while keeping all other PCs at $\alpha = 0$. The corresponding spatial maps for each of those PCs are shown on the right in Fig 3b, with each cell represented by a colored dot varying from purple $\alpha \approx -3$ to pink $\alpha \approx 0$ to yellow $\alpha \approx +3$. To assess the distribution of the latent features relative to the point of impact, we denoted the impact site for each sample with a yellow arrow.

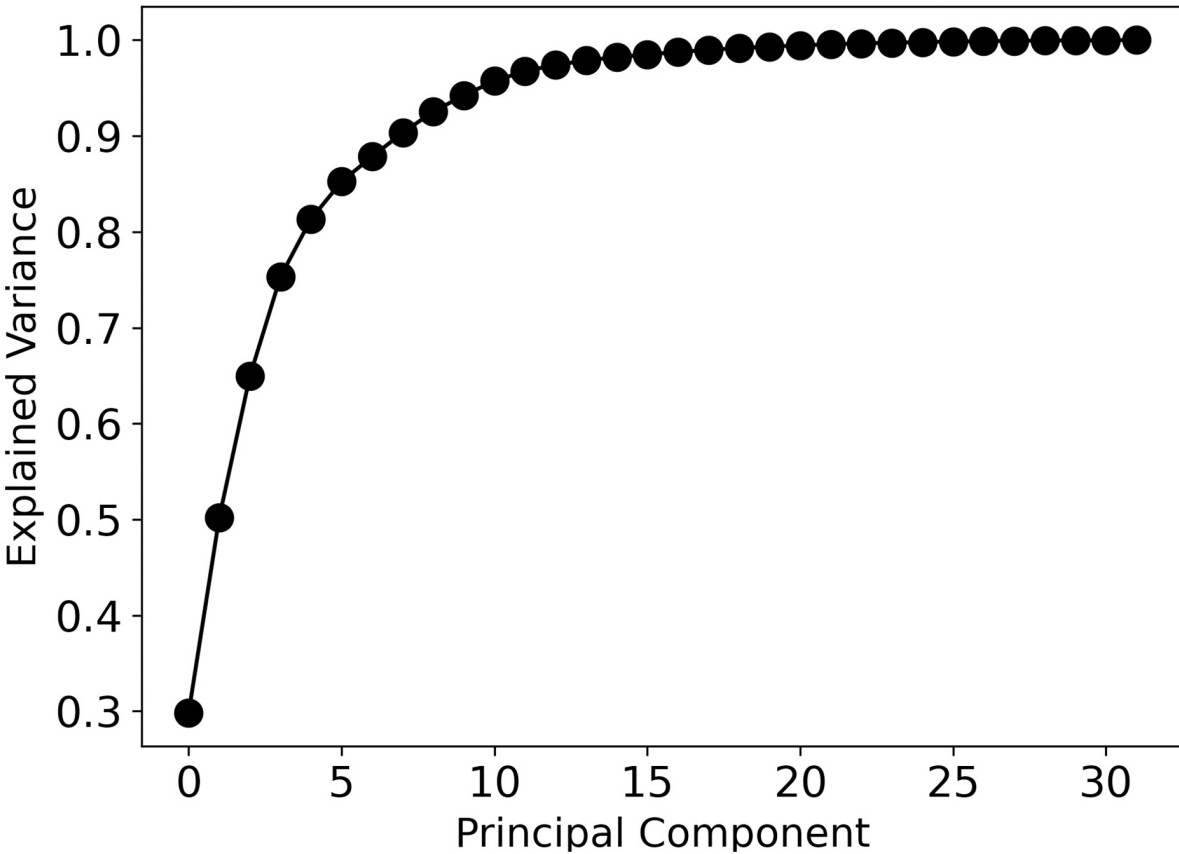

**Fig 2. Explained variance ratio for each principal component (PC).** The 32 PCs in this system are plotted. Up to 90% of the variance can be explained using only the first 8 PCs.

PC0 encodes cells that died right after impact (high nuclear membrane permeability, blue curve) at $\alpha = -3$ to cells with decaying but elevated calcium concentration (green curve) and decaying mitochondrial polarity (red curve) at $\alpha = +3$. This PC covers the spectrum between cell death due to impact and cells with slowly decaying function, as expected in an *ex vivo* setup. The spatial distribution of cells that exhibit the two distinct cellular behaviors encoded by PC0's extreme end can be clearly observed in the spatial map in Fig 3b. Notably, cells at the exact impact location (shown with orange arrows in Fig 3b) and some cells below the impact exhibit strongly negative values in this PC, whereas cells to the side of impact and on the control sample show strongly positive values in this PC.

PC1 encodes distinct cellular behaviors, associating negative values with a low calcium signal and high but diminishing mitochondrial polarity, while high values correlate with declining calcium signals and slightly elevated nuclear membrane permeability. Cells exhibiting high PC1 values are predominantly located at or below the impact site, whereas cells with low PC1 values are dispersed away from the impact site. In contrast, PC2 captures a different type of cell death/dysfunction at high values, characterized by a simultaneous decay in all three signals over time. This pattern is particularly pronounced in the dataset on the right, centered at and below the impact location, indicating a potential positive correlation between PC2 and strain, considering this dataset experienced the highest strain. Notably, negative PC2 values encode a subtle plateau within the calcium decay, a feature not captured in the preceding PCs. Moving

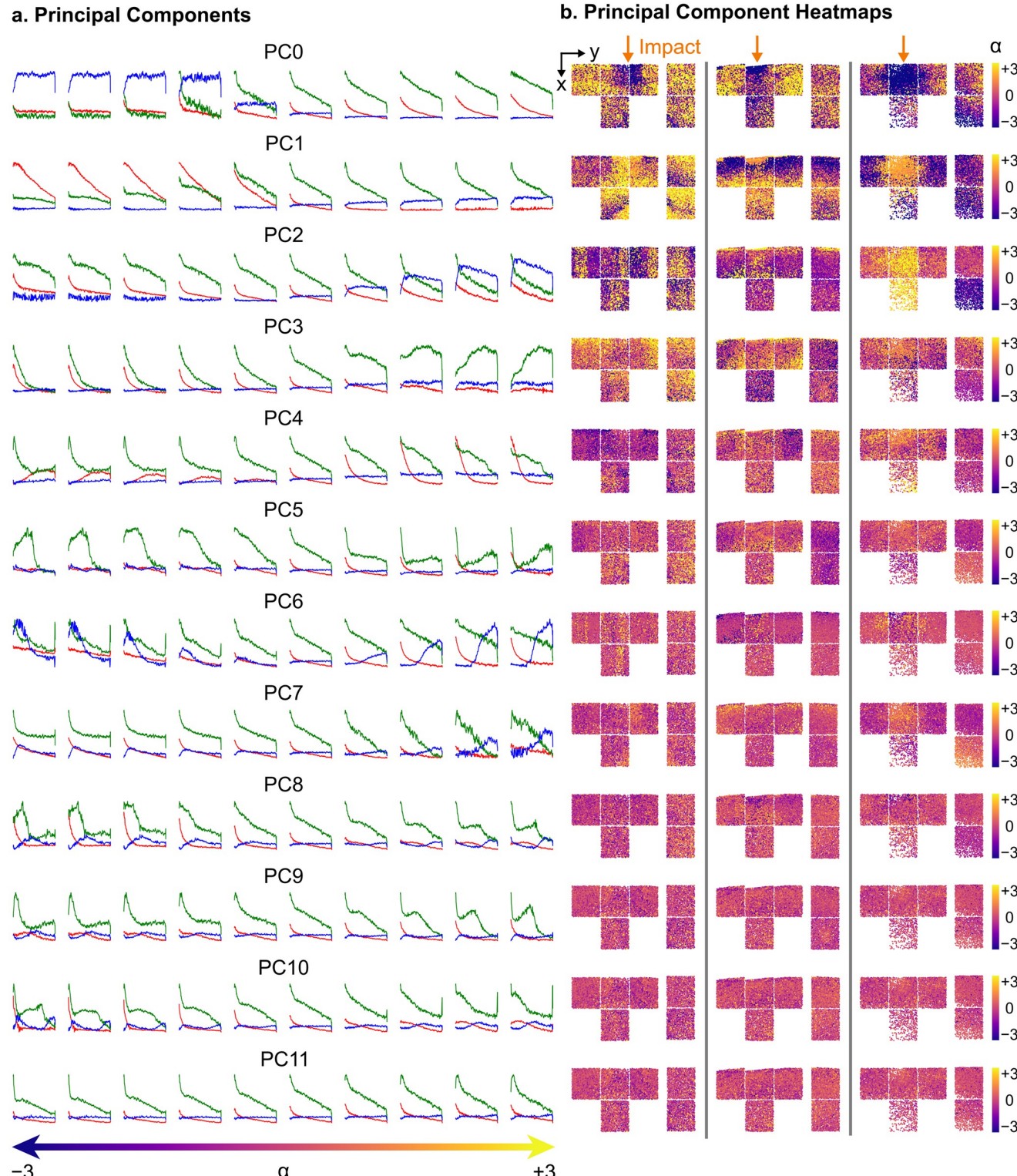

**Fig 3. Principal components and associated spatial heatmaps for three impact experiments.** Left: example reconstructed plots showing the variation along the first twelve principal components, varying from $\alpha = -3$ to $\alpha = 3$. Right: Spatial maps of all cells and their associated $\alpha$ values for each PC, where $\alpha$ varies from $-3$ (purple) to 3 (yellow). Impact location shown with orange arrow. The experiment in the rightmost column was subject to larger impact strain than the other two experiments (left and center).

to PC3, negative values capture a more rapid calcium decay, while positive values encode a decaying calcium activity with a late rise in activity, representing a distinct cellular behavior absent in previous PCs.

Successive PCs contributed progressively less to describing the explained variance ratio, evident in the more uniform colormaps ($\alpha \approx 0$). Noteworthy features captured by the VAE include increased calcium concentration followed by cell death at different time points for PC5, PC9, and PC10, as well as late cell death at unique time points for PC6, PC7, and PC8. While these behaviors were less prevalent in the observed cell population, the VAE successfully learned them. Examining the relative importance and spatial distributions of these PCs revealed that, while the dominant behavior of cell death was apparent, the cell death process could be broadly categorized into three distinct modes: 1) Instantaneous cell death—characterized by highly elevated nuclear membrane permeability signals with minimal other signals, 2) high $Ca^{2+}$ cell death/dysfunction—marked by somewhat elevated nuclear membrane permeability signals along with elevated calcium signals, and 3) cell death/dysfunction—featuring highly elevated but decaying nuclear membrane permeability signals along with decaying other signals. Importantly, these categories align with three of the most distinctive features used in manual labeling in [11], swiftly distinguished by the VAE without necessitating time-consuming analysis.

## Clustering post-impact data into distinct behavior phenotypes

We next used the latent features to differentiate cells with different post-impact responses. As each cell behavior is parameterized by 32-dimensional VAE latent features (mean vector, $\vec{\mu}$ and diagonal covariance vector, $\vec{\sigma}$) that describe a 32-dimensional Gaussian distribution, we took an information geometric approach [29] in quantifying the dissimilarity between two distributions. In information geometry, divergences quantify differences between distributions. Unlike conventional metrics, divergences need not be symmetric or satisfy the triangle inequality. Multivariate Gaussian distributions fall within the parametric set of distributions known as the exponential family, encompassing widely used distributions like the Bernoulli and Chi-square distributions [28]. The Kullback-Liebler divergence, also known as the relative entropy,

$$\mathrm{KL}(P|Q) = \int p(x) \log p(x)/q(x) dx \tag{3}$$

where $P$ and $Q$ are continuous random variables, $p$ and $q$ are the probability densities of $P$ and $Q$, was shown to be the canonical divergence of the exponential family [28, 29]. In our analysis, we utilized the symmetrized version $D_{sKL} = \frac{1}{2}(\mathrm{KL}(P|Q) + \mathrm{KL}(Q|P))$ [27] to measure the similarity between the cell's latent probability distributions.

We performed agglomerative hierarchical clustering based on the $D_{sKL}$ between latent features to cluster our cells, where similar cells are grouped together iteratively until a predefined distance threshold is reached. As an illustration, Fig 4a shows a cell dataset's dendrogram. A predefined distance threshold is represented as a black line, whose intersection with the tree produces fourteen clusters shown in Fig 5. To demonstrate the splitting process of the clusters, the shaded gray box highlights one branch of the cluster tree and is expanded in Fig 4b.

At three clusters, cluster A largely comprises cells with decaying calcium concentration and decaying mitochondrial polarity with lower overall intensity. As we decrease the cutoff threshold, cluster A splits into cluster B, which has a higher mitochondrial polarity signal and a flat calcium concentration, and cluster C, which looks largely similar to cluster A. As the cutoff threshold is decreased again, cluster B split into D and C. Cluster C captures dead/damaged cells with elevated and decaying nuclear membrane permeability and calcium concentration,

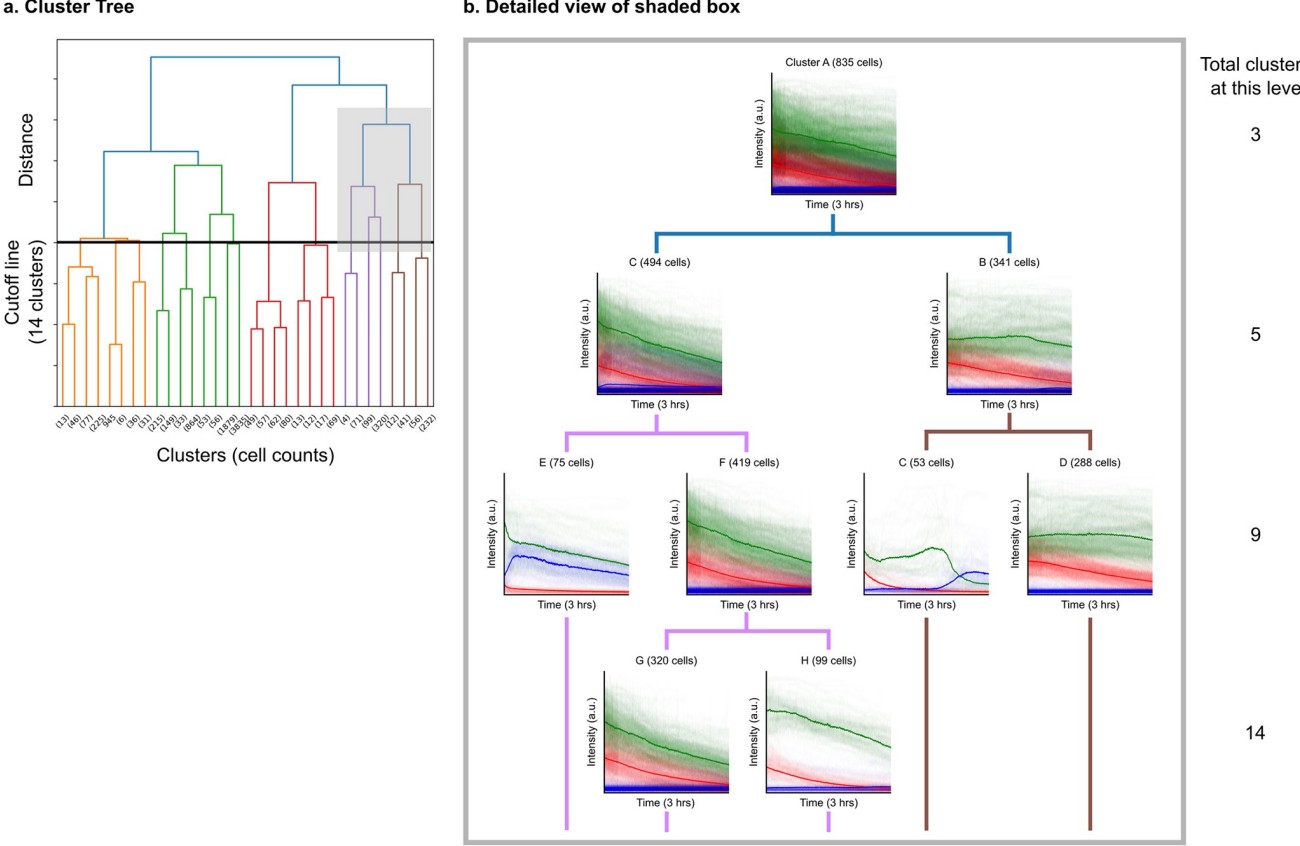

**Fig 4. Example branching of the hierarchical clustering tree.** Left: Overall agglomerative hierarchical clustering, with example cutoff line for 14 clusters shown in black. Each vertical line is a cluster. The x-axis shows cell counts and the y-axis represents distance between the clusters represented by the symmetrized Kullback-Liebner (sKL) divergence. Right: Expanded view of gray shaded box. Clusters break down into smaller groups with more detail when the total number of clusters is increased. Bolded lines represent cluster averages, and thin lines represent individual cells within a cluster. Red represents mitochondrial polarity, green calcium concentration, and blue nuclear membrane permeability (cell death). Four levels are shown, showing the specificity of clusters when there are 3, 5, 9, and 14 total clusters. Letters are used to name clusters for ease of identification and have no specific meaning.

whereas Cluster D again looks similar to its parent cluster. On the other hand, Cluster C splits into clusters E and F. Cluster E pulls out cells with calcium transient at roughly two hours into imaging, after which the cell dies, while Cluster F looks similar to its parent cluster. As the cutoff threshold is decreased further, cluster F split into two clusters (H and G) where the calcium concentration and mitochondrial polarity decay but have different overall intensity values.

To aid readers in interpreting these clusters, we have provided Table 1. Within our dataset, cells within the predominant cluster, labeled as cluster 1, consistently exhibited low values in both mitochondrial polarity decay and calcium levels, accompanied by minimal nuclear membrane permeability signals. These cells were found throughout the sample. Cells within clusters 2 through 9, on the other hand, uniformly displayed low nuclear membrane permeability but demonstrated varied patterns in terms of calcium levels and mitochondrial polarity decay. The ability to differentiate between subtle differences in the decay profile allows us to more effectively distinguish between groups of viable cells, which will help to further our understanding of beneficial processes in subsequent experiments. This methodology safeguards against the introduction of human biases when categorizing cell behaviors and has the potential to unveil subtle cellular behaviors that could be overlooked through manual classification.

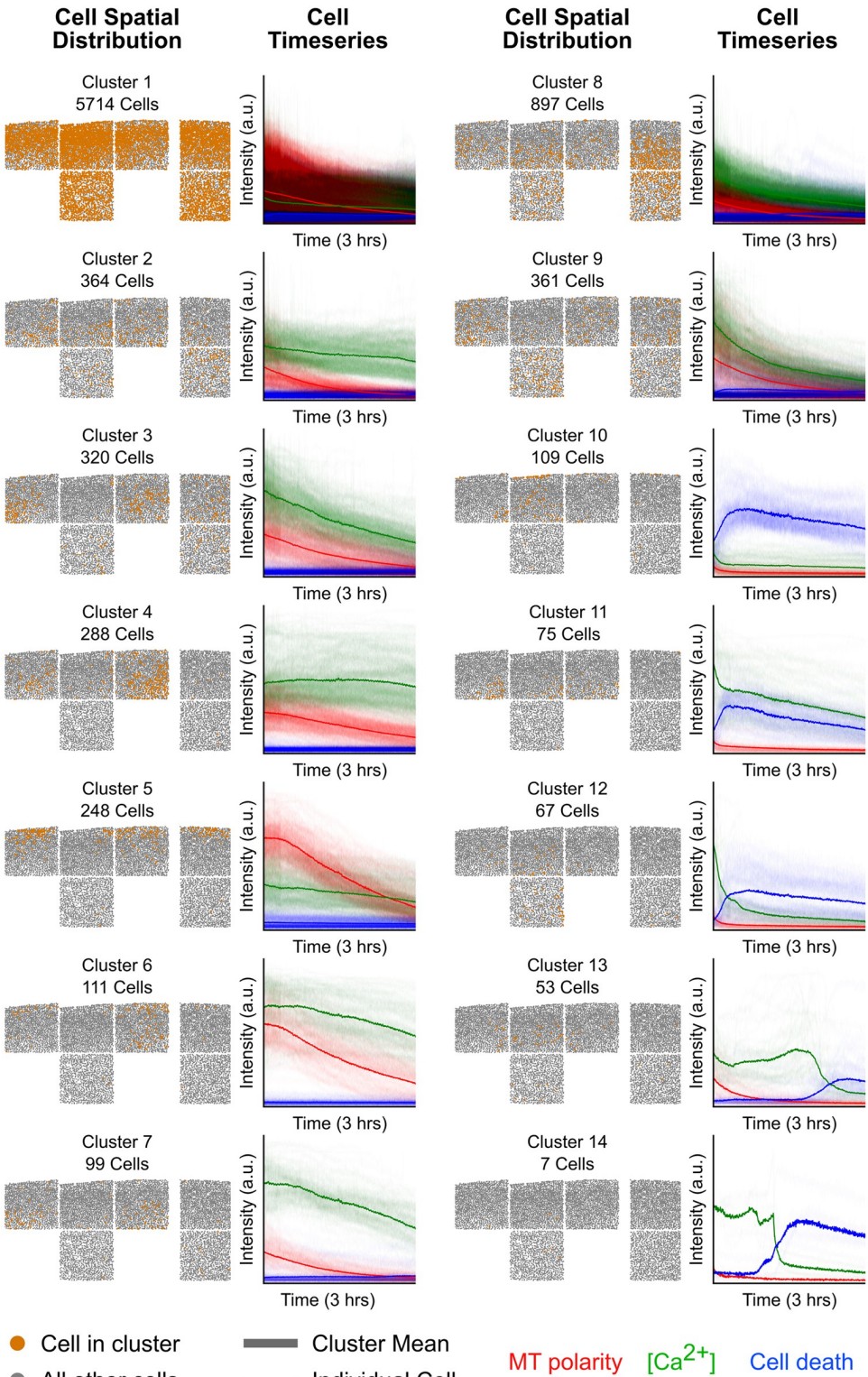

**Fig 5. Cell behavior clusters generated by the VAE.** All fourteen clusters from one example data set, organized in two columns. Each cluster is shown in three panels: the spatial distribution of cells within the cluster (cells represented as orange dots), the timeseries plots of cells within the cluster (with means as bolded lines), and the UMAP distribution of cells within the cluster (cells represented as orange dots). The clusters do not have a specific ordering.

**Table 1. Descriptions of cell behaviors for each cluster in Fig 5.**

| Cluster | Description | # of cells |
|---|---|---|
| 1 | Decaying MT polarity, low calcium concentration, low and medium nuclear membrane permeability | 5714 |
| 2 | Decaying MT polarity, medium non-decaying calcium concentration, no nuclear membrane permeability | 364 |
| 3 | Decaying MT polarity, medium decaying calcium concentration, no nuclear membrane permeability | 320 |
| 4 | Decaying MT polarity, medium non-decaying calcium concentration, no nuclear membrane permeability | 288 |
| 5 | High decaying MT polarity, medium non-decaying calcium concentration, no nuclear membrane permeability | 248 |
| 6 | High decaying MT polarity, high decaying calcium concentration, no nuclear membrane permeability | 111 |
| 7 | Decaying MT polarity, high decaying calcium concentration, no nuclear membrane permeability | 99 |
| 8 | Decaying MT polarity, medium decaying calcium concentration, low nuclear membrane permeability | 897 |
| 9 | Decaying MT polarity, medium decaying calcium concentration, no nuclear membrane permeability | 361 |
| 10 | No MT polarity, no calcium concentration, high nuclear membrane permeability | 109 |
| 11 | No MT polarity, medium decaying calcium concentration, high nuclear membrane permeability | 75 |
| 12 | No MT polarity, low decaying calcium concentration, high nuclear membrane permeability | 67 |
| 13 | No MT polarity, late calcium concentration rise and fall ($\sim$ 2 hrs), late rise in nuclear membrane permeability | 53 |
| 14 | No MT polarity, calcium concentration fall ($\sim$ 1.5 hrs), late rise in nuclear membrane permeability | 7 |

Cells in clusters 10 through 14 exhibited distinct patterns of increased nuclear membrane permeability. In Cluster 10, cells were characterized by a significant initial spike in nuclear membrane permeability at the onset of imaging, followed by a subsequent decay with no other discernible signals. These cells were primarily located at and just below the impact site. Cluster 11 comprised cells with a similar initial peak in nuclear membrane permeability but also displayed an elevated but decaying calcium concentration and were more broadly distributed around the impact site. Cluster 12 featured cells that underwent early death in the imaging process, marked by a considerably faster decay in calcium levels and less pronounced nuclear membrane permeability decay scattered around the impacted sample. Meanwhile, Clusters 13 and 14 presented a notably delayed cell death following a calcium transient, with Cluster 13 cells dying approximately 2 hours into the imaging session and Cluster 14 cells at around 1.3 hours.

## Generating and validating hypotheses with VAE generated clusters

To illustrate the utility of our VAE in hypothesis generation and validation, we present three examples that leverage the generated outputs. We first wondered about the potential predictive power of immediate calcium signaling (1 minute post-impact) on late-stage cell behavior (3 hours post-impact). To elucidate whether immediate calcium signaling predicts late-stage cell behavior, we examined the corresponding impact calcium signaling from the post-impact clusters. As our experimental protocol only recorded immediate calcium signaling at the impact

site, we conducted hierarchical clustering of the post-impact data in that specific region. We found a distance threshold that yielded 9 clusters is effective in capturing distinct cellular behaviors within each cluster, as shown in the long-term imaging cluster columns of Fig 6. We observed that the long-term imaging clusters are associated with initial calcium signals that

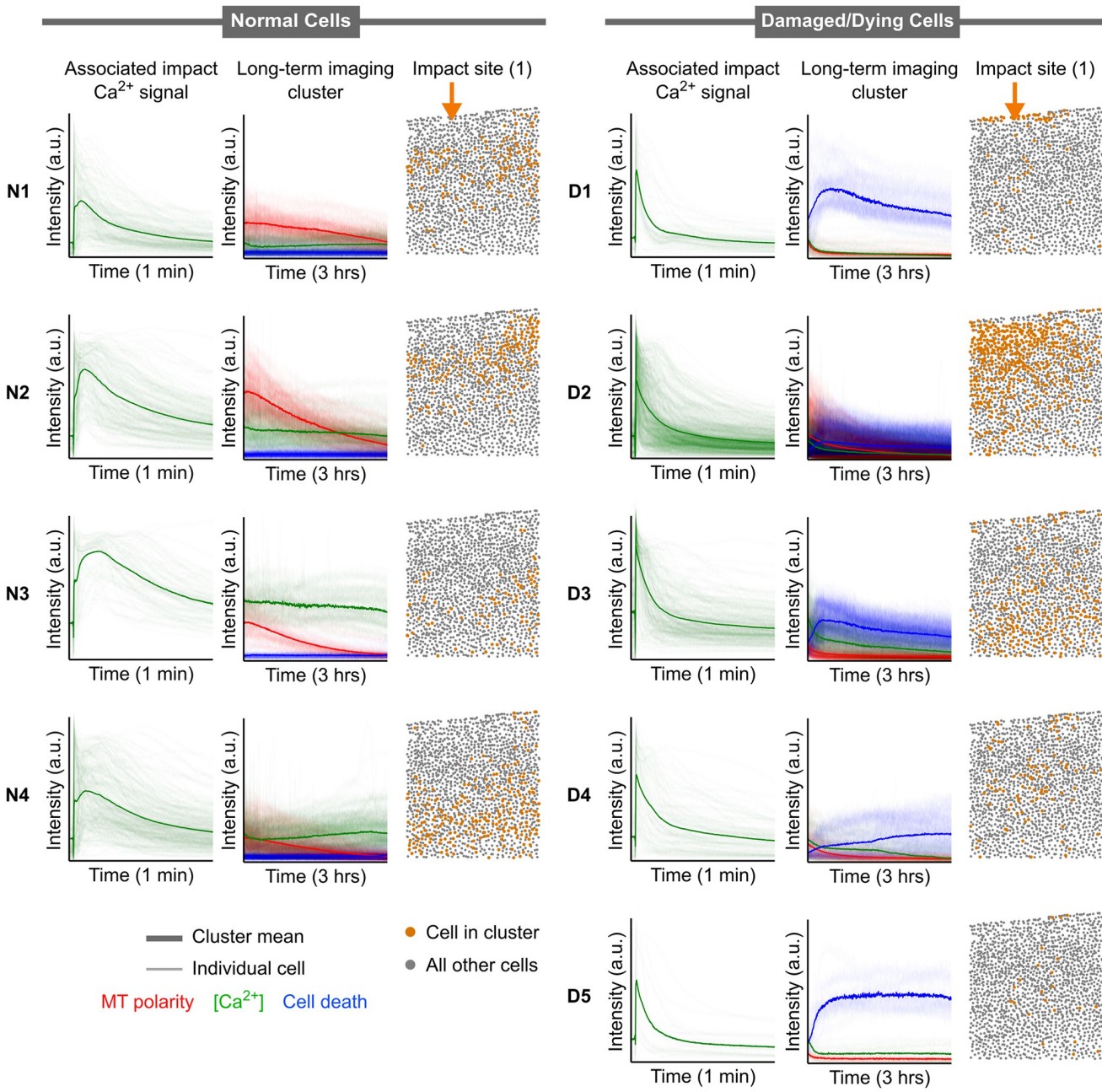

**Fig 6. Impact calcium signatures associated with VAE-produced clusters.** Differentiation between healthy cells and damaged or dying cells can be seen from clustering on post-impact long term data. Each cluster has three panels: the timeseries of the impact imaging for each cluster (1 minute) which includes only calcium concentration, the timeseries of all cells in each cluster for longer-term imaging (3 hours) which includes all three color channels (VAE-generated cluster), and the spatial distribution of cells in said cluster which are shown as orange dots. In timeseries plots, bolded lines represent means while thin lines represent individual cells within the cluster. The exact impact location is indicated with an orange arrow. Left: clusters showing functional/healthy cells. Right clusters showing damaged/dying cells.

can broadly categorized into those exhibiting plateaus of calcium concentration after impact and those with sharp peaks that decay over time. Notably, cells displaying higher overall calcium intensity and faster decay of the calcium signal were associated with increased nuclear membrane permeability in the hours following impact (Fig 6 right). These cells were predominantly localized directly below the impact site. Conversely, cells that demonstrated broad peaks or plateaus of calcium concentration after impact (reaching lower overall intensity compared to peaks in the previous categories) were more often associated with cells that did not die after impact (Fig 6 left). This observation led us to hypothesize that the sharpness of the $Ca^{2+}$ peak from impact-induced trauma contributes to cell death.

To test this hypothesis, we performed Principle Components Analysis (PCA) on the first minute of the $Ca^{2+}$ time series data. We then extracted the first principle component, which captures 80.33% variation of the data and can account for the range of observed peak shapes. This procedure enables us to parameterize the $Ca^{2+}$ data with a single number reflecting the sharpness measure (Fig 7a). We then utilized the clusters in Fig 5 to classify cells with high and low nuclear membrane permeability, where a high value is representative of dead cells and a low value is representative of viable cells. We found that the cells with high nuclear membrane permeability are associated with high values of the sharpness measure and are distributionally distinct from cells with low nuclear membrane permeability, as exemplified in Fig 7b.

Second, we tested whether late rises in nuclear membrane permeability, such as those observed in Fig 5 Cluster 13 and 14, were correlated to the proximity to the impact site. Our null hypothesis is that the timing of late cell death (where high nuclear membrane permeability is taken to indicate cell death) is independent of total strain, implying no correlation between the distance from the point of impact and the time at which cells die. To do this, we utilized the clusters generated via the agglomerative clustering technique to identify clusters with delayed rise in nuclear membrane permeability. We found cells exhibiting this characteristic in two cartilage samples (Fig 8a) and computed the cells' delayed rise time. We found no obvious relationship between the cell's delayed rise time and their distance from the point of impact (Fig 8b). Consequently, our findings failed to reject our null hypothesis. We observed a significantly higher incidence of delayed cell death in impacted samples than in control

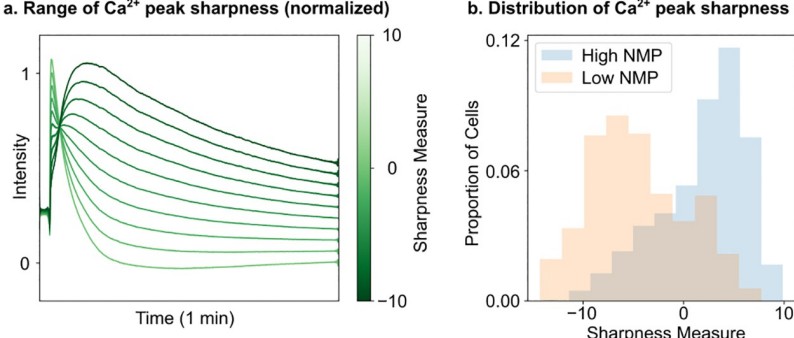

**Fig 7. Sharpness of the calcium peak is associated with different outcomes in cell fate.** a) PCA performed on the normalized calcium concentrations of cells at the impact site (position 1) for 1 minutes after impact. A range of sharpness can be observed, from immediate sharp peaks and fast decay (high sharpness measure) to later broad peaks with slow decay (low sharpness measure). b) Distribution of cells with high NMP (dead cell) or low NMP (viable cell). The clusters were generated with a hierarchical clustering technique utilizing the symmetrized KL divergence between the cell's latent distribution. Clusters with dead cells (blue-shaded) have much higher sharpness than clusters with viable cells (orange-shaded). Two sample Kolmogorov-Smirnov test was performed for statistical differences between the two distributions, $p = 2.22 \times 10^{-15}$.

samples, indicative of the delayed cell death mechanism related to impact-induced biochemical mechanism.

Finally, we investigated the relationship between cells with high mitochondrial polarity and their proximity to the point of impact. Previous work has demonstrated that mitochondrial depolarization precedes cell death and that higher local strain is associated with cell death [30, 31]. We made use of three articular cartilage samples experiencing different impact strengths. We identified clusters with high and low mitochondrial polarity by utilizing clusters generated from the agglomerative clustering technique described above. Clusters featuring cells exhibiting high TMRM signals were categorized as indicative of polarized mitochondria, whereas clusters housing cells displaying low TMRM signals were classified as representative of depolarized mitochondria. We found that clusters with high mitochondrial polarity were located further away from the impact site than clusters with low mitochondrial polarity (Fig 9), rejecting the null hypothesis that high mitochondrial polarity is independent of its location from the impact site. Future work involving strain measurement should enable the investigation of differences in cell response in relation to compression and shear.

## Discussion

The utilization of Variational Autoencoders (VAE) for analyzing spatiotemporal cellular behavior data at the tissue scale showcases the effectiveness of this unsupervised learning approach. The VAE framework enhances our analytical capabilities by efficiently processing high-dimensional data in multi-channel fluorescence microscopy videos with minimal user input. This accelerated processing enables rapid iteration through experimental inputs, allowing us to test hypotheses by manipulating individual factors and observing changes to clusters or identifying varying spatial distributions.

The VAE method exhibits numerous advantages over traditional manual labeling systems or supervised machine learning. Notably, it excels in handling noisy data, eliminating the need for extensive background correction. Additionally, once input parameters are configured, no additional work is required to produce meaningful results. Furthermore, the VAE

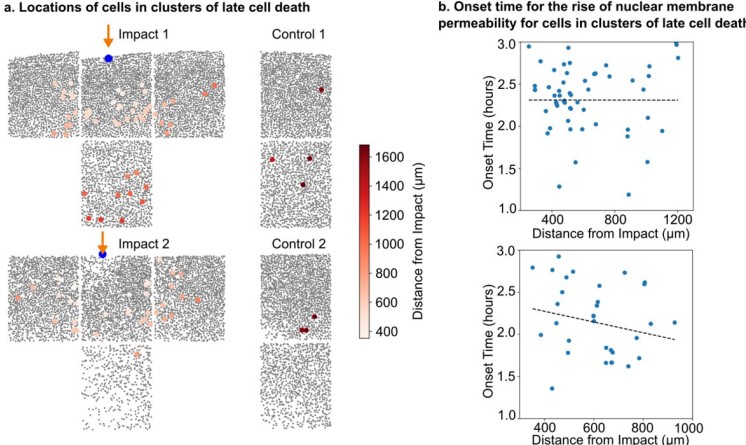

**Fig 8. Late cell death is present in impacted regions of cartilage but not associated with distance from impact.** a) Locations of cells showing late cell death (> 1 hour after impact) for two different impacts. Few cells in the control sample show late cell death, indicating that this behavior is likely related to impact. Colorbar illustrates the distance of each cell to impact site. b) The time that a cell reaches its maximum nuclear membrane permeability (indication of cell death) in these clusters is not associated with Euclidean distance from impact (F-test, top: $p = 0.996$, bottom: $p = 0.285$).

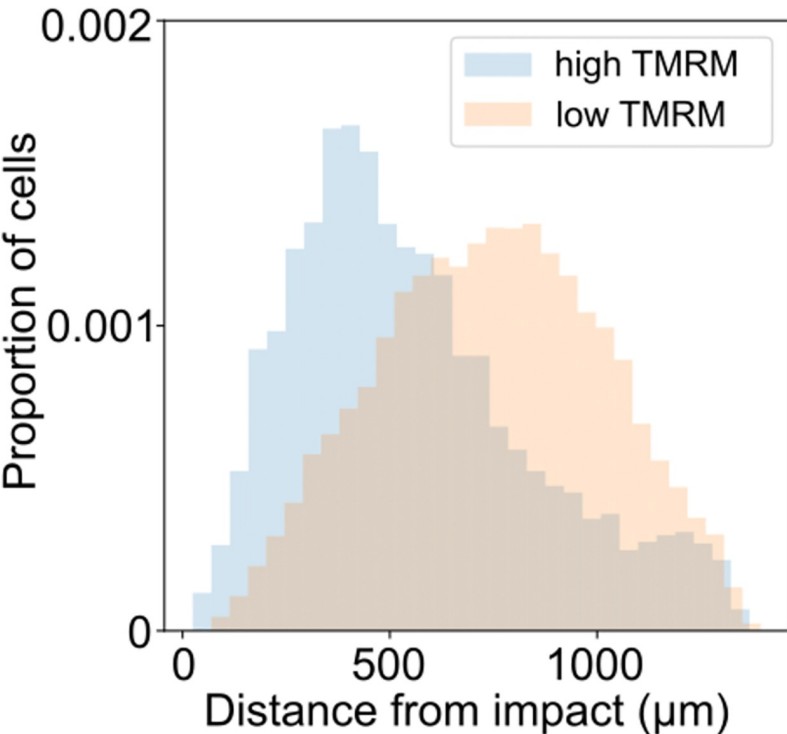

**Fig 9. Mitochondria depolarization occurs in impact region.** Distribution of cells with high TMRM (high polarized mitochondria) or low TMRM (low polarized mitochondria). The clusters were generated with a hierarchical clustering technique utilizing the symmetrized KL divergence between the cell's latent distribution. Clusters with low-polarized mitochondria (blue-shaded) are closer to the point of impact than clusters with high-polarized mitochondria (orange-shaded). Two sample Kolmogorov-Smirnov test was performed for statistical differences between the two distributions, $p = 1.43 \times 10^{-268}$.

accommodates multi-channel and non-normalized data, considering both absolute intensity and relationships between channels in determining cellular behaviors.

This implementation of the VAE is readily adaptable to analyze any tissue system where behaviors of cells are captured over time, not only from fluorescence microscopy. The method is not confined to any specific geometry, has no limitations on dataset size, and can work with time series data of any dimension. The VAE can be trained on any number of samples while still producing unique clusters for individual samples, enabling us to quickly look at sample-to-sample variation while simultaneously gaining a sense of the dominant, overarching behavioral themes with the principal components.

Use of the VAE to understand temporal patterns of chondrocyte signaling yielded new insights into the process of cartilage damage after injury. For example, the kinetics of calcium transport into chondrocytes was shown to be strongly associated with an increase in nuclear membrane permeability, an established indicator of cell death. Specifically, cells experiencing very sharp calcium transients were far more likely to die than those experiencing slower calcium transients. The connection between calcium influx and chondrocyte death is well established [32]. Further, recent data suggest that stretch-activated calcium channels such as TRPV4 and Piezo1 regulate calcium response to physiologic and non-physiologic loading, respectively [33–35]. Despite the abundance of interest in this topic, prior to the current work, there was no quantitative information on how the kinetics of calcium transients was related to

ultimate cell fate. Notably, the distinct phenotypes identified by the VAE enabled the determination that the sharpness of calcium peaks was a strong predictor of cell death, demonstrating the power of this technique to give important new insights into mechanisms of cell signaling.

One limitation of the current study is that in its present form, the VAE cannot capture spontaneous spontaneous calcium spikes, as is evident from the long-tail error distribution (Fig 1e right). There are a variety of computational techniques that could be used to analyze the spiking activities. For example, one could construct two network modules that jointly learn short-time scale spiking activities and long-timescale cellular profiles. From our observation, the spiking activities are relatively sparse; one may need to construct relevant features describing the spontaneous calcium spike for training. Another interesting limitation is the interpretability of the VAE latent features. For example, it is currently not obvious if each VAE dimension captures a particular biological process. In part, this may reflect the fact that typically, such latent spaces may have intrinsic curvatures [36], making it difficult to interpret distances properly. We are currently working on differential geometry methods to address these issues.

Finally, the VAE can be combined with other methods to comprehensively analyze cellular data in a tissue system, specifically with recently developed techniques focusing on spatially-resolved cell data [37–40]. Additionally, it is compatible with our previously published supervised learning/decision tree method [11], STRAINS, and works as a complementary technique for high-throughput analysis of spatiotemporal cell data. Further, samples are preserved after imaging, allowing for post-imaging analysis of gene expression, protein synthesis, cell metabolic activity, etc. By combining our VAE method with these analyses, a fuller picture of tissue-scale behaviors can be created.

## Author Contributions

**Conceptualization:** Michelle L. Delco, Lawrence J. Bonassar, Itai Cohen.

**Data curation:** Jingyang Zheng, Han Kheng Teoh.

**Formal analysis:** Jingyang Zheng, Han Kheng Teoh.

**Funding acquisition:** Michelle L. Delco, Lawrence J. Bonassar, Itai Cohen.

**Investigation:** Jingyang Zheng, Han Kheng Teoh, Itai Cohen.

**Methodology:** Jingyang Zheng, Han Kheng Teoh, Michelle L. Delco, Lawrence J. Bonassar, Itai Cohen.

**Project administration:** Michelle L. Delco, Lawrence J. Bonassar, Itai Cohen.

**Resources:** Michelle L. Delco, Lawrence J. Bonassar, Itai Cohen.

**Software:** Han Kheng Teoh.

**Supervision:** Michelle L. Delco, Lawrence J. Bonassar, Itai Cohen.

**Validation:** Michelle L. Delco, Lawrence J. Bonassar, Itai Cohen.

**Visualization:** Jingyang Zheng, Han Kheng Teoh.

**Writing – original draft:** Jingyang Zheng, Han Kheng Teoh.

**Writing – review & editing:** Jingyang Zheng, Han Kheng Teoh, Michelle L. Delco, Lawrence J. Bonassar, Itai Cohen.

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
