## [Decision Letter · Decision Letter 0]

7 Nov 2023

PONE-D-23-31622Application of a Variational Autoencoder for Clustering and Analyzing in situ Articular Cartilage Cellular Response to Mechanical StimuliPLOS ONE

Dear Dr. Teoh,

Thank you for submitting your manuscript to PLOS ONE. After careful consideration, we feel that it has merit but does not fully meet PLOS ONE’s publication criteria as it currently stands. Therefore, we invite you to submit a revised version of the manuscript that addresses the points raised during the review process.

We look forward to receiving your revised manuscript.

Kind regards,

Mohamed Yacin Sikkandar

Academic Editor

PLOS ONE

Journal Requirements:

"Funding: The work was supported by the NIH National Institute of Arthritis and Musculoskeletal and Skin Diseases, Contract: K08AR068470, R03AR075929, and The Harry M. Zweig Fund for Equine Research. This work was also supported by the NIH National Institute of Neurological Disorders and Stroke. Contract: R01NS116595. Additionally, this work was supported by the National Science Foundation grants DMR-1807602, CMMI 1927197, and BMMB-1536463. Lastly, this work made use of the Cornell Center for Materials Research Shared Facilities which are supported through the NSF MRSEC program (DMR-1719875)." 

Reviewers' comments:

Reviewer's Responses to Questions

**Comments to the Author**

1. Is the manuscript technically sound, and do the data support the conclusions?

Reviewer #1: Partly

Reviewer #2: Yes

2. Has the statistical analysis been performed appropriately and rigorously? 

Reviewer #1: Yes

Reviewer #2: Yes

3. Have the authors made all data underlying the findings in their manuscript fully available?

Reviewer #1: Yes

Reviewer #2: Yes

4. Is the manuscript presented in an intelligible fashion and written in standard English?

Reviewer #1: Yes

Reviewer #2: Yes

5. Review Comments to the Author

Reviewer #1: Authors have used VAE for analysing and classifying cartilage cellular response. Study is interesting and informative but not very clear.

Following are the comments to improve the quality and clarity of the manuscript:

1. Thorough editing is required - In this paper, 'introduction' section is immediately followed by 'results' and then explanation about the technologies adopted. Why is it so?

2. Under the section 'results' only outline of the work is discussed

3. Uniformity in 'speech style' may be followed - passive voice is normally followed to specify the action rather the person or entity performing the action.

4. More emphasis is required on the constructed VAE architecture with the function and encoding techniques adopted

5. 'Distance' is used as a measure to quantify the dissimilarity. What type of 'distance' is used and why?

6. Please present the quantified output, major findings and validation clearly

Reviewer #2: Is there any reason to discuss the results section first followed by methodology?

Previous use cases of VAE have been cited. Please highlight the specific biological context in which VAE is applied, and what is its intended outcome or objective within this context?

Also, please provide limitations of the present study and future directions for research.

6. PLOS authors have the option to publish the peer review history of their article (what does this mean?). If published, this will include your full peer review and any attached files.

Reviewer #1: No

Reviewer #2: No

---

## [Author Response · Author response to Decision Letter 0]

5 Dec 2023

*** We have attached a pdf containing the responses to reviewers comments in the submission system. 

We thank you for your time and valuable feedback. We were glad to see that Reviewer 1 found our work interesting and informative. Reviewer 2 had a number of mainly editorial suggestions that they wanted addressed. In particular, both reviewers suggested reordering the sections to have the methods sections before the results section. In response to both reviewers' editorial comments, we have diligently addressed their suggestions in this response letter, and corresponding revisions have been made to the manuscript.

Outlined below are the major modifications we have made:

- We have reorganized the manuscript to adhere to the standard format of Introduction-Methods-Results-Discussion.

- A significant expansion of the Results section has been undertaken to provide a more comprehensive discussion of our findings. 

- The Method section now includes a more detailed discussion of the VAE architecture employed in our study.

- The Discussion section has been expanded to encompass a more thorough discussion of the current study's limitations and potential avenues for future research (as requested by reviewer 2).

- Finally, we have modified the manuscript to address confusions.

Detailed response to Reviewer 1

1. Thorough editing is required - In this paper, 'introduction' section is immediately followed by 'results' and then explanation about the technologies adopted. Why is it so?

Response:

We have changed the sections order from Introduction-Results-Discussion-Method to Introduction-Method-Results-Discussion

2. Under the section 'results' only outline of the work is discussed

Response:

We thoroughly went through the manuscript again to make appropriate changes to better orient the readers. We highlight some of the major changes to the result section below. 

 - Line 64-76 were modified to include a detailed description data generation protocol

 - Line 102- 133 were rewritten to expand on the discussion of the PCA analysis significantly

 - Lines 155-206 were rewritten to expand on the clusters found via hierarchical clustering significantly

 - Lines 207-232 were rewritten to significantly expand on the discussion of the hypothesis testing using VAE generated clusters

3. Uniformity in 'speech style' may be followed - passive voice is normally followed to specify the action rather than the person or entity performing the action.

Response:

We thank the reviewer for the feedback. We have modified the manuscript according to this recommendation to the best of our ability. 

4. More emphasis is required on the constructed VAE architecture with the function and encoding techniques adopted

Response:

We thank the reviewer for the feedback. We have included a more detailed description of the constructed VAE architecture in the Method section, reproduced below.

 - Line 346-377 were modified to emphasize the constructed VAE architecture: 

The encoder is composed of three 1D convolution layers featuring a kernel size of 3, a stride of 2, and a padding size of 1.The input and output channels are configured as follows: i) $(3,4)$, ii) $(4,8)$, and iii) $(8,16)$. Following the convolutional layers is a fully connected layer with input and output channels set at $(1504,256)$. The resulting output is then directed into two branches of fully connected layers, where each branch further comprises two fully connected layers with input and output channels specified as i) $(256,128)$ and ii) $(128, 32)$. These two branches are responsible for generating the latent means and variance vectors. All encoder layers employ a tanh activation function except for the final output layer. Additionally, batch normalization is applied to the data as it passes through the convolutional layers.

The decoder is comprised of three fully connected layers, where the input and output channels are defined as i) $(32,128)$, ii) $(128,256)$, and iii) $(256,1504)$, respectively. The output from the last fully connected layer is reshaped to attain dimensions of $(B,16,94)$, with $B$ representing the batch size. Subsequently, this reshaped output is passed through three transposed convolution layers, each featuring a kernel size of 3, a stride of 2, and a padding size of 1. The output padding parameters are adjusted to ensure that the final output of the transposed convolutional layers restores the original time series dimension. Similar to the encoder architecture, a tanh activation function is applied throughout, with the exception of the final output layer. Moreover, latent vectors undergo batch normalization as they pass through the transposed convolutional layers.

To train the Variational Autoencoder (VAE), the time series data $x$ is passed through the encoder, yielding latent means $\\mu_{\\phi}(x)$ and latent diagonal covariance $\\sigma_{\\phi}(x)$ vectors that define a $32$-dimensional normal distribution, denoted as $q_{\\phi}(z|x) = \\mathcal{N}(\\mu_{\\phi}(x),\\sigma_{\\phi}(x))$, where $\\phi$ denotes the weights and biases of the encoder. Subsequently, a sample $z$ is drawn from the $32$-dimensional Gaussian and propagated through the decoder, parameterized by weights and biases $\\theta$, to produce a reconstructed intensity profile. The optimization of the encoder and decoder's weights and biases $\\phi, \\theta$ involves maximizing the log-likelihood of generating real data, $\\log p_{\\theta}(x)$, while simultaneously minimizing the information loss when the encoder distribution $q_{\\phi}(z|x)$ is used to approximate the true posterior distribution, $p_{\\theta}(z|x)$. The information loss can be quantified via the KL divergence, $D_{\\text{KL}}(q_{\\phi}(z|x)|p_{\\theta}(z|x))$.

The VAE loss function, denoted as $\\mathcal{L}_{\\text{VAE}}$, is formulated as:

\\textcolor{blue}{\\begin{equation}

 \\mathcal{L}_{\\text{VAE}} = \\log{p_{\\theta}(x)} - D_{\\text{KL}}(q_{\\phi}(z|x)||p_{\\theta}(z|x)).

\\end{equation}

In practice, this optimization is achieved by minimizing the evidence lower bound objective (ELBO),

\\begin{equation}

\\mathcal{L}_{\\text{ELBO} } = - \\mathbb{E}_{z\\sim

q_{\\phi}(z|x)} \\log p_{\\theta}(x|z) + D_{\\text{KL}}(q_{\\phi}(z|x)||p_{\\theta}(z))

\\end{equation}

which consists of two terms: 1) expected log-likelihood of the decoder distribution, which minimizes the prediction error of the reconstruction, and 2) a regularization term that seeks to minimize the difference between the encoder distribution $q_{\\theta}(z|x)$ and the prior distribution, $p(z)$. Here, we assume the prior distribution over the latent features $z$ to be unit Gaussian, $\\mathcal{N}(0,\\mathbb{I})$. 

5. 'Distance' is used as a measure to quantify the dissimilarity. What type of 'distance' is used and why?

Response:

The VAE encodes cellular profiles into two sets of latent features, which parameterize a 32-dimensional Gaussian distribution. To assess the dissimilarity between two multivariate Gaussian distributions, we adopted an information geometric approach. In information geometry, divergences quantify differences between distributions. Unlike conventional metrics, divergences need not be symmetric or satisfy the triangle inequality. Multivariate Gaussian distributions fall within the parametric set of distributions known as the exponential family, encompassing widely used distributions like the Bernoulli, Chi-square distributions, and others. The Kullback-Liebler divergence, also known as the relative entropy, 

\\begin{equation}

 \\text{KL}(P|Q) = \\int p(x) \\log{p(x)/q(x)} dx 

\\end{equation}

where $P$ and $Q$ are continuous random variables, $p$ and $q$ are the probability densities of $P$ and $Q$, was shown to be the canonical divergence of the exponential family, and in our analysis, we employed the symmetrized version to measure dissimilarity. We have modified the manuscript to include more details about the distance used, as described below.

Line 156-161 were modified to include a detailed explanation of the choice of distance used:

We next used the latent features to differentiate cells with different post-impact responses. As each cell behavior is parameterized by 32-dimensional VAE latent features ( mean vector, $\\vec{\\mu}$ and diagonal covariance vector, $\\vec{\\sigma}$ ) that describe a 32-dimensional Gaussian distribution, we took an information geometric approach \\cite{amari2000methods} in quantifying the dissimilarity between two distributions. In information geometry, divergences quantify differences between distributions. Unlike conventional metrics, divergences need not be symmetric or satisfy the triangle inequality. Multivariate Gaussian distributions fall within the parametric set of distributions known as the exponential family, encompassing widely used distributions like the Bernoulli and Chi-square distributions \\cite{nielsen2009statistical}. The Kullback-Liebler divergence, also known as the relative entropy,

\\begin{equation}

 \\text{KL}(P|Q) = \\int p(x) \\log{p(x)/q(x)} dx 

\\end{equation}

where $P$ and $Q$ are continuous random variables, $p$ and $q$ are the probability densities of $P$ and $Q$, was shown to be the canonical divergence of the exponential family \\cite{amari2000methods,nielsen2009statistical}. In our analysis, we utilized the symmetrized version $D_{sKL}= \\frac{1}{2}(\\text{KL}(P|Q)+\\text{KL}(Q|P))$ \\cite{teoh_visualizing_2020} to measure the similarity between the cell's latent probability distributions. We performed agglomerative hierarchical clustering based on the $D_{sKL}$ between latent features to cluster our cells, where similar cells are grouped together iteratively until a predefined distance threshold is reached.

6. Please present the quantified output, major findings, and validation clearly

Response:

We have worked to improve the readability of the results and discussion sections. Since we had to reorder the sections, we were not able to track individual sentence changes. 

Detailed response to Reviewer 2

1. Is there any reason to discuss the results section first, followed by methodology?

Response:

We have changed the sections order from Introduction-Results-Discussion-Method to Introduction-Method-Results-Discussion

2. Previous use cases of VAE have been cited. Please highlight the specific biological context in which VAE is applied, and what is its intended outcome or objective within this context?

Response:

We thank the reviewer for the feedback. In this paper, we aim to elucidate cellular behavior in cartilage tissue following impact-induced injury, which is crucial for gaining insights into tissue function in healthy and diseased states. As the traditional analysis pipeline necessitates manual processing of an overwhelming volume of data, the intended objective of using a VAE is to 

demonstrate its utility in rapidly iterating through experimental inputs and test hypotheses by changing individual factors and observing changes to clusters or identifying varying spatial distributions. Importantly, VAEs have thus far not been used to analyze this type of data in cartilage tissue. We have modified the introduction, reproduced below, to highlight better the biological context in which the VAE was applied and its intended objective.

 - Lines 46-52 were modified to present the objective of the paper more clearly:

This paper aims to demonstrate the utility of VAEs for analyzing large-scale cellular response in articular cartilage to mechanical stimuli - a process traditionally requiring a time-consuming analysis pipeline. We showcase a novel use of VAEs to accurately reconstruct temporal features of cellular behavior in cartilage and leverage latent features for phenotype identification immediately after injury. Additionally, we highlight the VAE's role in hypothesis generation and validation, providing a comprehensive understanding of cell response post-injury.

3. Also, please provide limitations of the present study and future directions for research.

Response:

We thank the reviewer for the feedback. One of the limitations of the current study is the inability of the constructed VAE to capture spontaneous calcium spikes, as exemplified in the long tail error distribution (Fig. 1e right). While the spiking activities can be analyzed separately with other computational techniques, in principle, one could envision constructing two network modules that jointly learn the short-time scale spiking activities and long-timescale cellular profile. This could potentially be achieved by decomposing the cellular profiles into two time series of short and long timescales. From our observation, the spiking activities are relatively sparse; one may need to construct relevant features describing the spontaneous calcium spike for training.

Another interesting limitation is the interpretability of the VAE latent features. For example, it is currently not obvious if each VAE dimension captures a particular biological process. In part, this may reflect the fact that typically, such latent spaces may have intrinsic curvatures \\cite{arvanitidis2017latent} making it difficult to interpret distances properly. We are currently working on differential geometry methods to address these issues. 

We have expanded the discussion section to include the limitations and future directions discussed above, reproduced below.

A new paragraph was added after Line 295 to discuss the limitation and future directions, reproduced below:

One limitation of the current study is that in its present form, the VAE cannot capture spontaneous calcium spikes, as is evident from the long-tail error distribution (Fig. 1e right). There are a variety of computational techniques that could be used to analyze the spiking activities. For example, one could construct two network modules that jointly learn short-time scale spiking activities and long-timescale cellular profiles. From our observation, the spiking activities are relatively sparse; one may need to construct relevant features describing the spontaneous calcium spike for training.

Another interesting limitation is the interpretability of the VAE latent features. For example, it is currently not obvious if each VAE dimension captures a particular biological process. In part, this may reflect the fact that typically, such latent spaces may have intrinsic curvatures \\cite{arvanitidis2017latent} making it difficult to interpret distances properly. We are currently working on differential geometry methods to address these issues.

---

## [Decision Letter · Decision Letter 1]

16 Jan 2024

Application of a Variational Autoencoder for Clustering and Analyzing in situ Articular Cartilage Cellular Response to Mechanical Stimuli

PONE-D-23-31622R1

Dear Dr. Teoh,

We’re pleased to inform you that your manuscript has been judged scientifically suitable for publication and will be formally accepted for publication once it meets all outstanding technical requirements.

Kind regards,

Mohamed Yacin Sikkandar

Academic Editor

PLOS ONE

Additional Editor Comments (optional):

Reviewers' comments:

Reviewer's Responses to Questions

**Comments to the Author**

1. If the authors have adequately addressed your comments raised in a previous round of review and you feel that this manuscript is now acceptable for publication, you may indicate that here to bypass the “Comments to the Author” section, enter your conflict of interest statement in the “Confidential to Editor” section, and submit your "Accept" recommendation.

Reviewer #1: All comments have been addressed

Reviewer #2: All comments have been addressed

2. Is the manuscript technically sound, and do the data support the conclusions?

Reviewer #1: Yes

Reviewer #2: Yes

3. Has the statistical analysis been performed appropriately and rigorously? 

Reviewer #1: Yes

Reviewer #2: Yes

4. Have the authors made all data underlying the findings in their manuscript fully available?

Reviewer #1: Yes

Reviewer #2: Yes

5. Is the manuscript presented in an intelligible fashion and written in standard English?

Reviewer #1: Yes

Reviewer #2: Yes

6. Review Comments to the Author

Reviewer #1: Quality of the figures could be improved.

Important result/outcome of the paper to be included in the abstract (one or two sentences)

Reviewer #2: (No Response)

7. PLOS authors have the option to publish the peer review history of their article (what does this mean?). If published, this will include your full peer review and any attached files.

Reviewer #1: No

Reviewer #2: No
